# An SPS-RS Technique for the Fabrication of SrMoO_4_ Powellite Mineral-like Ceramics for ^90^Sr Immobilization

**DOI:** 10.3390/ma16175838

**Published:** 2023-08-25

**Authors:** Anton A. Belov, Oleg O. Shichalin, Evgeniy K. Papynov, Igor Yu. Buravlev, Arseniy S. Portnyagin, Semen A. Azon, Alexander N. Fedorets, Anastasia A. Vornovskikh, Erhan S. Kolodeznikov, Ekaterina A. Gridasova, Anton Pogodaev, Nikolay B. Kondrikov, Yun Shi, Ivan G. Tananaev

**Affiliations:** 1Nuclear Technology Laboratory, Department of Nuclear Technology, Institute of High Technologies and Advanced Materials, Far Eastern Federal University, 10 Ajax Bay, Russky Island, 690922 Vladivostok, Russiapapynov@mail.ru (E.K.P.); buravlev.i@gmail.com (I.Y.B.); arsuha@gmail.com (A.S.P.); azon.sa@dvfu.ru (S.A.A.); fedorets.alexander@gmail.com (A.N.F.); vornovskikh_aa@dvfu.ru (A.A.V.); pogodaev.av@dvfu.ru (A.P.); kondrikov.nb@dvfu.ru (N.B.K.);; 2Shanghai Institute of Ceramics, Chinese Academy of Sciences, Shanghai 201899, China; 3Center of Materials Science and Optoelectronics Engineering, University of Chinese Academy of Sciences, Beijing 100049, China; 4Tananaev Institute of Chemistry and Technology of Rare Elements and Mineral Raw Materials, Kola Science Center, Russian Academy of Sciences, Akademgorodok, 26a, 184209 Apatity, Russia

**Keywords:** ceramics, strontium molybdate, powellite, radionuclides, radioactive waste management, solid-phase synthesis, SPS

## Abstract

This paper reports a method for the fabrication of mineral-like SrMoO_4_ ceramics with a powellite structure, which is promising for the immobilization of the high-energy ^90^Sr radioisotope. The reported method is based on the solid-phase “in situ” interaction between SrO and MoO_3_ oxides initiated under spark plasma sintering (SPS) conditions. Dilatometry, XRD, SEM, and EDX methods were used to investigate the consolidation dynamics, phase formation, and structural changes in the reactive powder blend and sintered ceramics. The temperature conditions for SrMoO_4_ formation under SPS were determined, yielding ceramics with a relative density of 84.0–96.3%, Vickers microhardness of 157–295 HV, and compressive strength of 54–331 MPa. Ceramic samples demonstrate a low Sr leaching rate of 10^−6^ g/cm^2^·day, indicating a rather high hydrolytic stability and meeting the requirements of GOST R 50926-96 imposed on solid radioactive wastes. The results presented here show a wide range of prospects for the application of ceramic matrixes with the mineral-like composition studied here to radioactive waste processing and radioisotope manufacturing.

## 1. Introduction

Mineral-like ceramics based on alkali-earth tungstates [1], molybdates [2,3,4], niobates [5], vanadates [6,7], etc., are actively studied as promising materials for the immobilization of radioactive wastes (RAW) in the form of solid matrixes for long-term storage. One of the best examples of these is the molybdate of a powellite structure that naturally occurs as CaMoO_4_, adopting a scheelite-type tetragonal phase (space group I 4 1/a, Z = 4). Molybdates are of particular interest due to their outstanding luminescence properties, which are applied in gas detectors [8,9], optical devices [10,11,12,13,14,15], and scintillation detectors [16,17,18,19]. Different synthesis and processing techniques lead to changes in structural properties, which affect the material properties and comprise a motivating factor in the investigation of the formation of SrMoO_4_. The interest in the field of immobilization comes from the structural variability that is due to the ability of molybdenum to reach several oxidation states (Mo^6+^, Mo^5+^, Mo^4+^, Mo^3+^); however, under oxidizing or neutral conditions, most molybdenum ions are hexavalent, existing as [MoO_4_]2-tetrahedra [20]. The structural flexibility of powellite allows the substance to tolerate large chemical substitutions including actinoid and lanthanide elements [2]. For example, Ca^2+^ cations in this structure can be partially or completely replaced with Sr^2+^, Ba^2+^, Cu^2+^, Mn^2+^, Cd^2+^, Pb^2+^, Cr^2+^, and Fe^2+^, and with rare-earth elements Nb^+2/+4^, Ta^+5^, and W^+6^ [20]. In other words, isostructural molybdate-like compounds can contain numerous elements, the isotopes of which are found in RAW [21]. Furthermore, Mo introduction into multicomponent highly active RAW at the level of 10 wt.% promotes the formation of the crystalline powellite phase, which has a positive effect on phase stability of the compound, as was previously shown based on the stability improvement in glasses [22].

Out of a large number of radionuclides that have to be immobilized, ^90^Sr is of particular importance. This radionuclide is characterized by high specific activity, a long half-life period, and the ability to activate heat generation. ^90^Sr is derived via β decay of ^90^Rb and its isotopes, followed by a similar decay pathway into ^90^Y. The latter one is also radioactive, with a half-life period of 64 h and β decays into ^90^Zr. In this respect, the choice of the matrix for ^90^Sr immobilization also accounts for the high liberated energy of the radioactive β decay of strontium itself (545.9 MeV) and its ^90^Y derivative (~2.28 MeV), which can irreversibly affect the physicochemical nature of the materials. Apart from that, strontium’s features, such as the ability to self-heat, gas migration, and high mobility in liquid media, are also put into consideration.

Several techniques are used to synthesize SrMoO_4_, including the Czochralski method [13], coprecipitation [7,18], traditional solid-phase reactions [12,23], sol-gel synthesis [9], hydrothermal synthesis [17], and microwave hydrothermal synthesis [15,16].

Among modern methods for the fabrication of RAW-immobilizing ceramics, one should pay special attention to spark plasma sintering (SPS) technology. The main principle is based on the rapid heating of the powder material in a vacuum by passing pulses of direct current through the graphite die with the sample while applying uniaxial pressure. SPS is characterized by a high heating rate, which allows us to shorten the fabrication time, reduce the cost of manufacturing, and improve the quality in comparison with the known counterparts. [24]. Successful examples of SPS application for solid-state matrix fabrication to immobilize RAW, including the ones with Sr, are in reports on studies of ceramics with structures of NZP, sinroc, garnet, whitlockite, perovskite, and others [25,26,27]. Additionally, the application of SPS technology was first implemented for the fabrication of radioisotope prototypes in the form of ionizing irradiation sources based on ceramic cores as active zones [28], some of which contain Sr [26,29].

In addition to that, a number of studies demonstrated high prospects for the implementation of SPS coupled with reactive sintering (SPS-RS) [30]. This approach is based on the “in situ” interaction between the components of the starting powder blend leading to the formation of ceramics composition different from that of reagents. Solid-state reaction between reagents is initiated under SPS conditions below their melting points [31]. This allows us to reduce the sintering temperature in the system, which in turn reduces the risk of sublimation of the sintered components, including the radioactive ones. Application of SPS-RS for fabrication of ceramics suitable for radionuclide immobilization is limited to a few reports studying chabazite [32], apatite [33], zirconate [34], zirconolite [35,36], scheelite [37], and feldspar [38]. SPS-RS fabrication of molybdate-based ceramics for radionuclide immobilization was not conducted before, to the best of our knowledge.

In this paper, the SPS-RS method is presented as a cumulative system in which simultaneous synthesis of the material and its ceramization takes place. In terms of synthesis, this method can be compared only with the method of solid-phase reaction followed by pressing and sintering [14]. The advantage of SPS technology becomes evident due to the shorter production cycle and one-stage production, and simultaneously applied pressure allows us to obtain ceramics with high parameters of mechanical strength.

It is important to note that many ceramics will be needed to immobilize large amounts of radioactive strontium. The final calculations of such ceramics are complex, as each specific case of radionuclide handling presents individual challenges. Industrial versions of SPS systems, such as the JXP series [39], can provide for the production of large quantities of ceramics. Such a system allows the production of samples with a diameter of up to 300 mm, and it can also be designed as a conveyor system. Such a system can significantly reduce the production time, which is especially important when dealing with radioactive waste.

Thus, this paper aims to investigate the solid-phase synthesis of SrMoO_4_ ceramics with a powellite structure via the SPS-RS fabrication pathway. This approach allows us to form solid-state matrixes with high exploitation characteristics, including hydrolytic stability, and which are suitable for the safe immobilization of Sr radionuclides.

## 2. Experimental

### 2.1. Reagents

The main precursors for the sample synthesis, namely, strontium oxide (SrO, 99.9%) and molybdenum oxide (MoO_3_, 99.9%), were purchased from Sigma Aldrich (St. Louis, MI, USA) and were used as received.

SrMoO_4_ ceramics was formed according to the following equation:SrO + MoO_3_ = SrMoO_4_

### 2.2. Reactive Mixture Preparation

The reactive mixture was prepared via ball milling of SrO (4.20 g) and MoO_3_ (5.82 g) oxides on a Tencan XQM-0.4A planetary mill at 870 rpm for 7 cycles of 15 min each, with a 15 min break in between.

### 2.3. SPS-RS Fabrication of SrMoO_4_ Ceramics

SrMoO_4_ ceramics were fabricated using the SPS-RS approach on an SPS-515S sintering setup (“Dr. Sinter·LAB^TM^”, Kyoto, Japan), according to the following scheme. The reactive mixture was placed into the graphite die (internal diameter 15.5 mm), prepressed at 20.7 MPa, transferred into a vacuum chamber (10^−5^ atm), and sintered. Heating was provided by unipolar low-voltage pulse current in on/off regime with pulse/pause periodicity 12/2 and duration 39.6/6.6 ms. SPS temperature was controlled via an optical pyrometer (low limit of detection was 650 °C), focused on a 5.5 mm deep gap in the middle of the outer die wall. To prevent the powder from baking onto the die inner walls and plunges as well as to ease the ceramic extraction, we used 200 µm graphite foil. The die was wrapped in a thermal-insulating fabric to reduce the heat loss. Diameter and height of the obtained matrixes in the form of a cylinder were 15.3 mm and 4–10 mm (depending on the sintering temperature), respectively.

Final sintering temperature spanned 800–1200 °C, and heating rate was changed in stages: 300 °C/min below 650 °C, and 50 °C/min above 650 °C, when the working range of the pyrometer was reached. Holding time at final temperature was 5 min, then samples were cooled down to room temperature for 30 min. Mechanical load during the whole process was kept constant at 24.5 MPa.

### 2.4. Ceramics Preparation Prior to Characterization

To remove graphite foil from the ceramic surface, coarse polishing was conducted on the first stage using silicon carbide sandpaper with the grain sizes US CAMI 80, 120, and 240 (Allied High Tech Products, Inc., Compton, CA, USA). Then, the fine polishing was performed using silicon carbide sandpaper with grain sizes US CAMI 400, 600, 800, and 1200, followed by polishing with colloidal diamond suspension with 9, 3, and 1 m and 0.04 µm sized particles (Allied High Tech Products, Inc., Compton, CA, USA) on a PRESI MECATECH 234 polishing machine (Eybens, France). 

### 2.5. Characterization Methods

Particle size distribution was determined on a particle size analyzer G3-ID manufactured by Malvern Instruments Ltd. (Malvern, UK). Scanning electron microscopy (SEM) was performed on a CrossBeam 1540 XB manufactured by Carl Zeiss (Jena, Germany), equipped with the add-on for energy-dispersive spectral analysis (EDX) by Bruker (Mannheim, Germany). XRD was carried out on a D8 Advance Bruker AXS (Mannheim, Germany) diffractometer. The grain size distributions and the average grain sizes were calculated by the linear intercept method using scanning electron microscopy (SEM) images [40]. At least 300 grains for each sample were analyzed for each measurement. Vickers microhardness (HV) was determined at 0.2 N load on the microhardness tester HMV-G-FA-D manufactured by Shimadzu (Kyoto, Japan). Compressive strength (σ_cs_) was evaluated on the tensile machine Autograph AG-X plus 100 kN manufactured by Shimadzu (Kyoto, Japan). Experimental density (ED) was measured by hydrostatic weighing on the balance Adventurer™ manufactured by OHAUS Corporation (Parsippany, NJ, USA). Relative density (RD) was found as a ratio of the experimental density (ED) measured via hydrostatic weighing to the theoretical density (TD).

Raman spectroscopy was carried out with an automated confocal micro-Raman setup (NTEGRA Spectra II, NT-MDT) equipped with a grating type spectrometer (M522, Solar Laser Systems) and a CCD-camera (i-Dus, Andor Technologies, Belfast, UK). Raman scattered from the isolated hemispherical NPs was excited by unpolarized fiber-coupled CW laser radiation (633 nm pump wavelengths) focused on the sample surface with a dry microscope objective (NA = 0.7; 100x Mitutoyo Apo).

Hydrolytic stability of matrixes was estimated based on desalination rate of strontium under long-term contact (30 days) with distilled water (pH 6.8) at room temperature (25 °C) in static conditions according to a well-known Russian Government Standard (GOST R 52126-2003), closely related to the ANSI/ANS—American National Standards Institute/American Nuclear Society 2019 (ANSI/ANS 16.1), which was updated according to the older procedure recommended by IAEA (ISO 6961:1982). Strontium ion concentrations were determined by inductively coupled plasma atomic emission spectrometry (ICP-MS) on an iCAP 7600 Duo spectrometer (Thermo Scientific, Waltham, MA, USA, 2013).

## 3. Results and Discussion

Fabrication of mineral-like ceramics with a powellite structure was based on the initiation of the “in situ” interaction between the oxides within the green body in the moment of SPS consolidation according to the following equation:SrO + MoO_3_ = SrMoO_4_(1)

According to particle size analysis (Figure 1a), the initial oxide blend is characterized by a wide bimodal fractional distribution in the range 0.1–10 µm, with mean sizes 0.2 and 2 µm corresponding to the two modes. Particle size data are confirmed by SEM images, showing that the reactive powder blend possesses polydisperse size composition (Figure 1b), with the small-sized fraction consisting of needle-shaped particles. Large particles are relatively homogeneously distributed within the small particle fraction.

Based on the temporal and thermal evolution of SPS consolidation (Figure 2a,b), it was found that sintering proceeds in two stages regardless of the final sintering temperature. The first stage encompasses mechanical densification of powder particles and their partial deformation, rearrangement, and packing under constantly applied pressure. The second densification stage at 580–650 °C is a result of chemical interaction yielding the SrMoO_4_ phase and thermal consolidation of particles forming a dense compact. This stage manifests in the active atomic diffusion along the interparticle contacts, viscous flow, and particle plastic deformation, leading to overall material compaction. The sample sintered at 1200 °C observed a third densification stage, likely caused by the sublimation of molybdenum oxide at 1155 °C in accordance with references [41].

According to XRD (Figure 3), the reactive interaction within the oxide blend leading to SrMoO_4_ formation under SPS proceeds successfully at all studied temperatures in the range 800–1200 °C. The intensity of diffraction maxima increases with the temperature. However, at 1200 °C, SrMoO_4_ starts to decompose back into oxides due to the sublimation of MoO_3_ occurring at 1155 °C, according to the previously reported data [41]. Pronounced decomposition can be a result of the vacuum employed during sintering, which promotes the removal of sublimated molybdenum oxide and shifts the equilibrium towards the reverse reaction. The presented data indicate the isomorphism of the CaMoO_4_-SrMoO_4_ structure and the possibility of Ca^2+^ substitution by Sr^2+^. The theoretical possibility of isomorphic substitution is proved by Goldschmidt’s law, which agrees on the following points:Ion sizes should differ by no more than 10–15%. The ionic radius of Ca^2+^ is 1.04 Å and the ionic radius of Sr^2+^ is 1.2 Å, resulting in a difference of 14%.The difference in electronegativity is less than 0.4. The electronegativity of Ca^2+^ is 1.00 and that of Sr^2+^ is 0.95, resulting in a difference of 0.05.

As can be seen, the conditions of isomorphism are fulfilled.

The structure of strontium molybdate has also been investigated by Raman spectroscopy (Figure 4). Raman spectra of molybdates include internal and external vibrational modes. Raman spectra of sintered SrMoO_4_ ceramic sample powders with ten observed and labeled vibrational modes are shown in Figure 4. The internal modes ν1, ν2, ν3, and ν4 are at 328, 365/366, 382, 796, 844, and 887 cm^−1^. The external modes are observed around 93–139 cm^−1^ and the free-spinning modes are observed at 182/179 cm^−1^, which is in agreement with the literature data.

SEM images reveal that the morphology of the powder particles of the starting reactive mixture significantly changes under SPS (Figure 5). The structure of the formed ceramics is presented by well-faceted grains. An increase in the reactive sintering temperature induces dramatic grain growth. Samples sintered at 800–1200 °C are characterized by the presence of macroporous formations. On the other hand, porosity diminishes at higher sintering temperatures, owing to the grain growth indicated above. The surface of the ceramics sintered at 1100–1200 °C approaches a monolith.

EDX data revealed that the distribution of the main elements on the shears of the analyzed surface is uniform. When comparing the obtained data with the literature sources [10,12,14], the similar morphology of particles as well as their dimensionality can be observed. 

The analysis of the morphology indicates that the samples obtained at sintering temperatures of 800–900 °C are characterized by a narrow grain size distribution with a predominance of grains of the smallest fractions (up to 2 μm) (Figure 6). The maximum size does not exceed 10 μm. The average grain sizes (d_AV_) for 800 °C and 900 °C samples are 2.6 and 2.9 μm, respectively. Starting at 1000 °C, a lognormal grain size distribution begins to form, and each subsequent 100 °C increase in SPS temperature leads to a twofold increase in average grain size. The grain distribution of the sample obtained at 1100 °C is well described by the lognormal function; the coefficient of determination is R^2^ = 0.968 and d_AV_ = 9.7 μm. The broadening of the distribution at 1200 °C (R^2^ = 0.895) indicates recrystallization processes in the system due to phase decomposition into the initial oxides. Maximum grain size reaches 54 μm.

The physicochemical characteristics of the ceramics, namely, relative density (84.0–96.3%), compressive strength (54–331 MPa), and Vickers microhardness (157–295 HV), increase with the sintering temperature (Figure 7a). A scatter plot of Vickers microhardness allows us to assess the mechanical microheterogeneity of the material (Figure 7b). According to the data, we found that each sample observed some variation in microhardness values caused by anisotropy of the studied characteristics in local parts of the sample. The sample sintered at 1100 °C observed increased heterogeneity in terms of microhardness, which is likely caused by molybdenum oxide sublimation. However, the sample sintered at 1200 °C exhibited much higher values of microhardness, probably due to the formation of separate phases of SrO and MoO_3_, which act as alloying agents strengthening the ceramic particles. Also, the effect of consolidation at 1200 °C is maximal compared to lower temperatures, yielding ceramics of higher mechanical strength.

As-prepared SrMoO_4_ ceramics were subjected to hydrolytic stability tests in terms of strontium discharge. The strontium leaching rate for all samples was as low as 10^−6^ g/cm^2^·day (Figure 8). The ceramic sample obtained at 1200 °C was characterized by a higher strontium leaching rate, even compared to the sample sintered at 800 °C. This effect is caused by the change in phase composition due to molybdenum oxide sublimation, proven by XRD above (Figure 3). It is noteworthy that all leaching rate values meet the requirements of ISO 6961:1982 and GOST R 50926-96, indicating the high hydrolytic stability of ceramics.

## 4. Conclusions

We realized the spark plasma sintering–reactive sintering strategy to fabricate SrMoO_4_ mineral-like ceramics adopting a powellite structure, which is a promising candidate for the immobilization of the high-energy ^90^Sr isotope. “In situ” consolidation dynamics of a SrO/MoO_3_ reactive powder blend were studied for the case of SPS. Dilatometry, XRD, SEM, and EDX methods were used to identify the main stages of reactive powder blend densification related to mechanical compaction and solid-state processes of sintering under thermal heating. Phase formation and structural and compositional changes of the formed ceramics were studied. The decomposition of the SrMoO_4_ phase into starting oxides above 1100 °C was revealed. Sintering was found to cause active grain growth, leading to the formation of a nonporous polydisperse ceramic structure. Higher sintering temperatures increase the relative porosity (84.0–96.3%), compressive strength (54–331 MPa), and Vickers microhardness (157–295 HV), with the sample obtained at 1200 °C attaining the best properties so far. All fabricated ceramics demonstrated low leaching rates of 10^−6^ g/cm^2^·day, indicating high hydrolytic stability that meets the requirements of GOST R 50926-96 and ISO 6961:1982 imposed on highly radioactive solid wastes.

## Figures and Tables

**Figure 1 materials-16-05838-f001:**
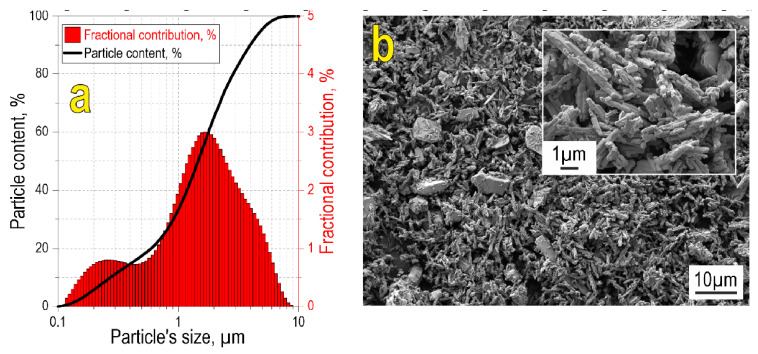
Particle size distribution (**a**) and SEM image (**b**) of the initial reactive powder blend. The inset in (**b**) shows the SEM image of the initial blend at higher magnification.

**Figure 2 materials-16-05838-f002:**
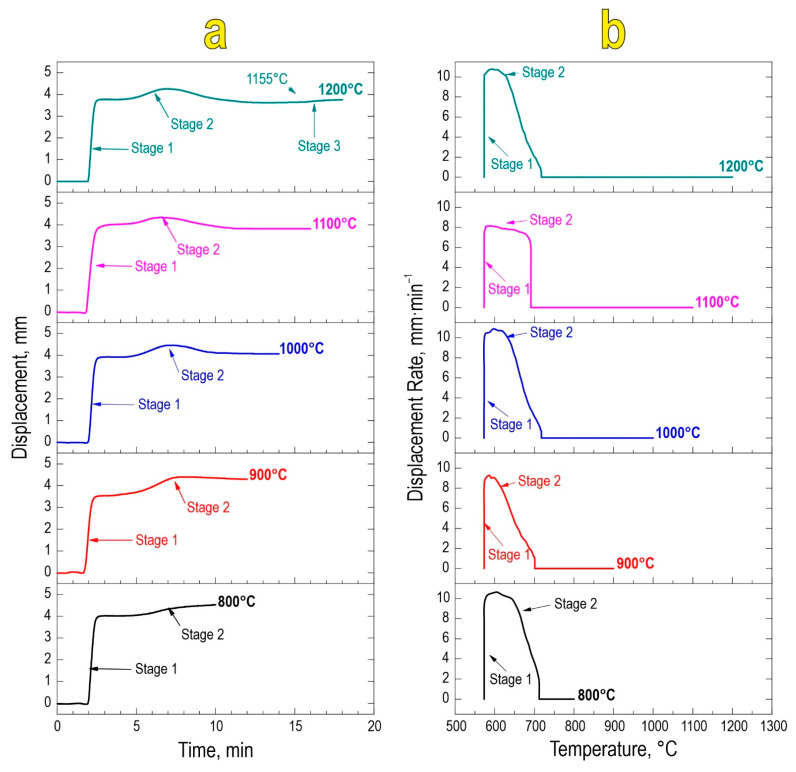
Consolidation dynamics of the reactive powder blend during SPS-RS, expressed as densification rate vs. time (**a**) and temperature (**b**).

**Figure 3 materials-16-05838-f003:**
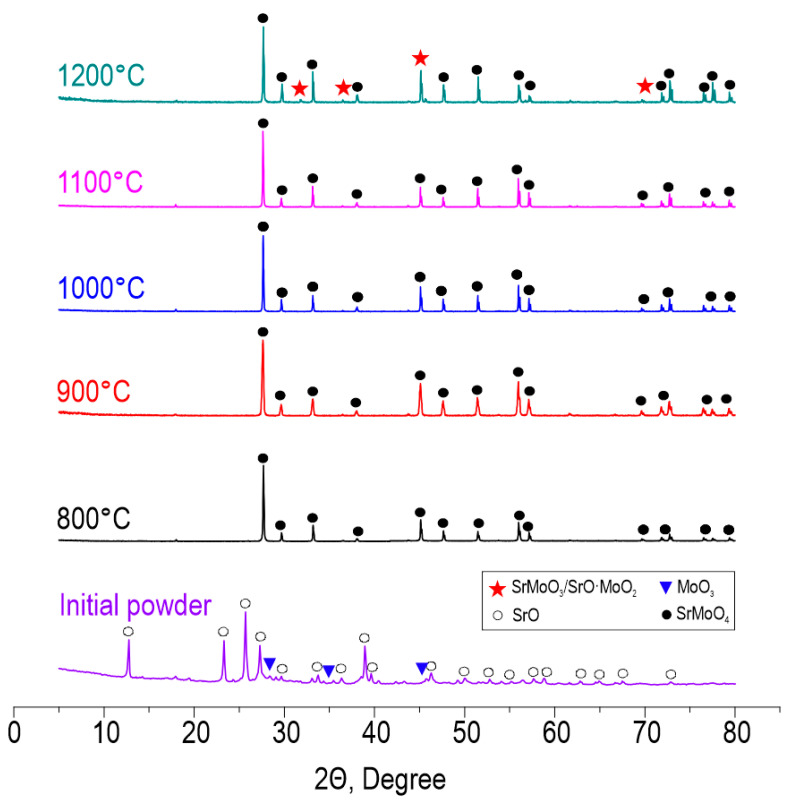
XRD patterns of the initial reactive powder blend and ceramic samples derived thereof at various SPS-RS temperatures.

**Figure 4 materials-16-05838-f004:**
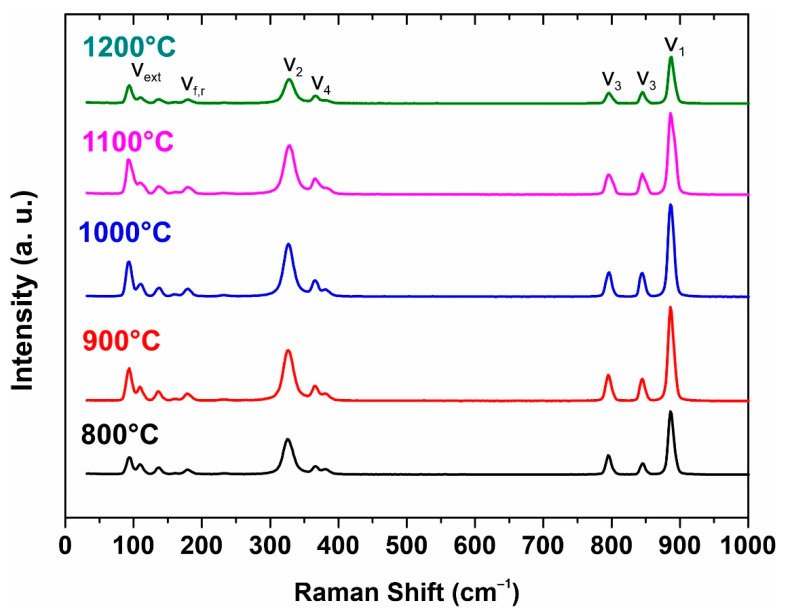
Raman spectra from initial powder to 1200 °C sample.

**Figure 5 materials-16-05838-f005:**
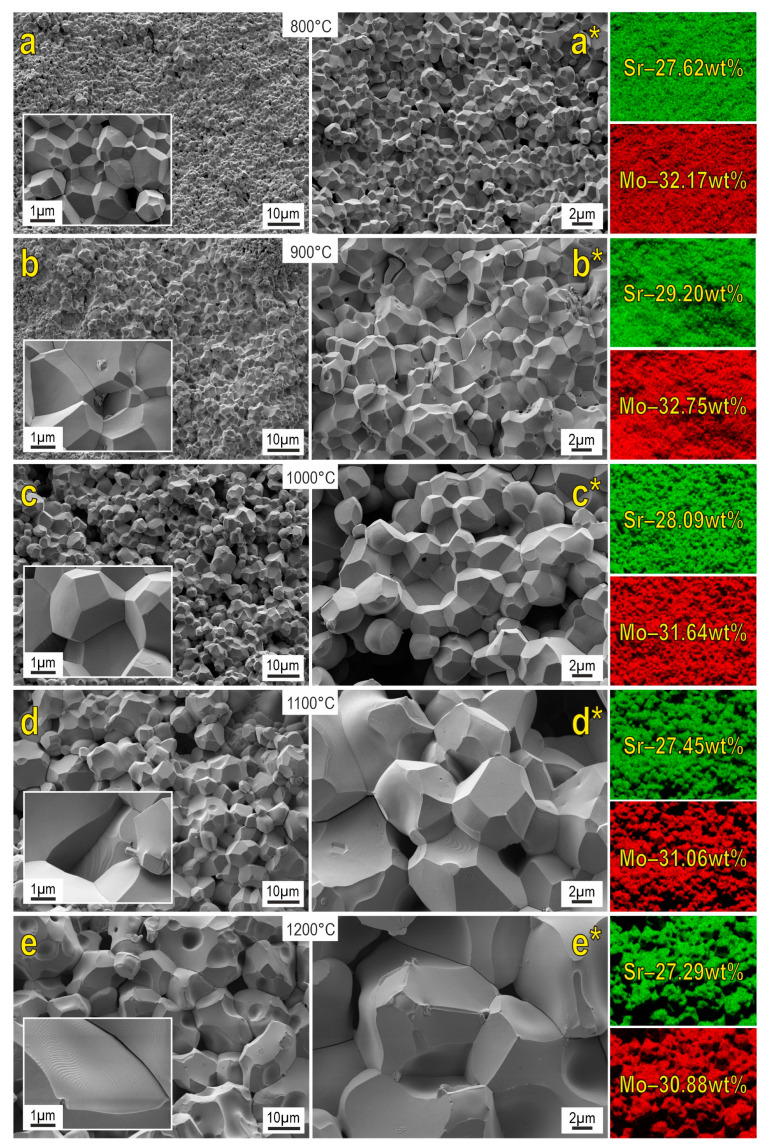
SEM images and EDX analysis of SrMoO_4_ ceramics sintered at various SPS-RS temperatures.

**Figure 6 materials-16-05838-f006:**
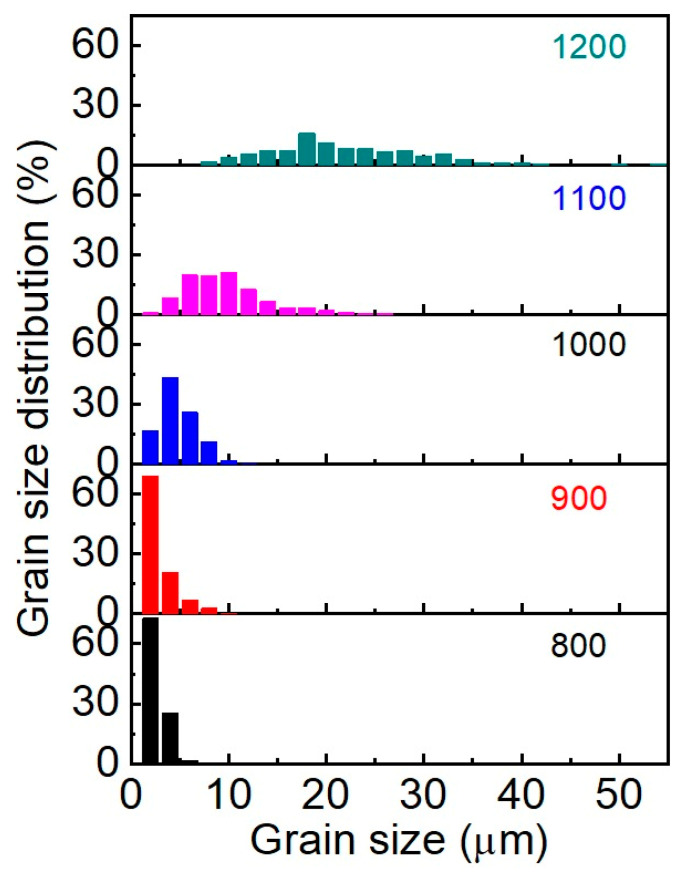
Grain size distribution of SrMoO4 SPS samples at 800–1200 °C.

**Figure 7 materials-16-05838-f007:**
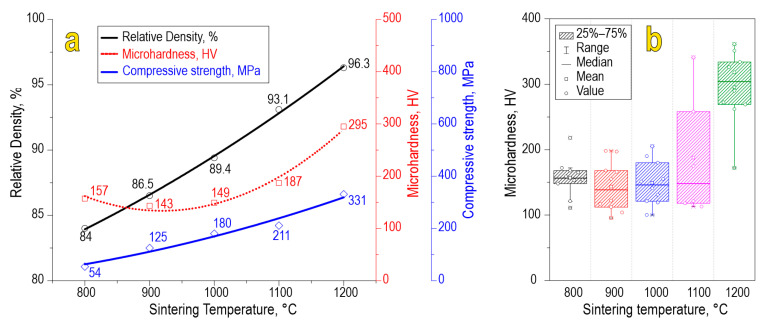
Physicochemical characteristics (**a**) and scatter plot of Vickers microhardness (**b**) for SrMoO_4_ ceramics sintered at various SPS-RS temperatures.

**Figure 8 materials-16-05838-f008:**
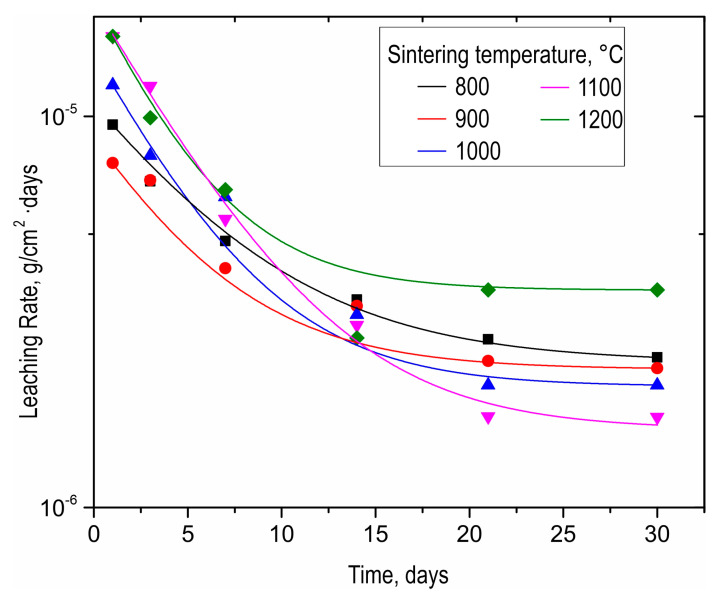
Leaching rate of strontium from SrMoO_4_ ceramics sintered at various SPS-RS temperatures during long-term (30 days) contact with distilled water.

## Data Availability

There are no databases or archives. All obtained results are displayed in the publication.

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
