# Peer review of "An SPS-RS Technique for the Fabrication of SrMoO4 Powellite Mineral-like Ceramics for 90Sr Immobilization"

_materials, 2023, doi:10.3390/ma16175838_

Round 1
Reviewer 1 Report
Referee report on manuscript “SPS-RS technique for SrMoO4 powellite mineral-like ceramic fabrication for 90Sr immobilization”
This is an interesting article, it can be certainly recommended for publication after clarifying some uncertainties.
1. Introduction. First sentence. It is important to note that all these compounds are even more important as scintillator materials for different application in high energy physics, medicine and other detector applications. See, for example:
Millers, D.; Grigorjeva, L.; Chernov, S.; Popov, A.; Lecoq, P.; Auffray, E. The temperature dependence of scintillation parameters in PbWO4 crystals. Phys. Status Solidi B 1997, 203, 585–589.
Jung, J.-Y. Luminescent Color-Adjustable Europium and Terbium Co-Doped Strontium Molybdate Phosphors Synthesized at Room Temperature Applied to Flexible Composite for LED Filter. Crystals 2022, 12, 552. https://doi.org/10.3390/cryst12040552
Srivastava, A. M., Brik, M. G., Beers, W. W., Ma, C. G., Piasecki, M., & Cohen, W. E. (2023). Intensity of the Eu3+ hypersensitive transition in isostructural phosphate and vanadate compounds. Journal of Luminescence, 257, 119709.
This is important to attract a wider readership and visibility of this work,
2. In the introduction, it is important to note how much this ceramic is required for 90Sr immobilization and whether the preparation technique SPS-RS under consideration will cope with this?
3. Whether this will affect the use of this ceramic, its subsequent radiation damage via point defect production? Especially since it is well known that under irradiation point defects are formed in all compounds (SrO, MoO3 and SrMO4-type).
4. Figure 4. It would be useful to perform also Raman measurements
5. How will porosity affect the declared application?
6. The list of references should be according to the MDPI standards.
In general, the manuscript is interesting and can be recommended for publication after constructive reflection on the above comments.
Author Response
Re1: “SPS-RS technique for SrMoO4 powellite mineral-like ceramic fabrication for 90Sr immobilization” by Belov A.A. and et al. (Manuscript Number: materials-2527932)
Dear, Dear Editors, Dear Reviewers,
We deeply appreciate the time you spent reviewing our paper and the valuable recommendations you made. All the comments are taken into account and corresponding changes are made to the manuscript’s body text. Detailed point-by-point answers are presented below.
Best regards,
Oleg Shichalin, Researcher, Ph.D.
Reviewer #1 comments:
Referee report on manuscript “SPS-RS technique for SrMoO4 powellite mineral-like ceramic fabrication for 90Sr immobilization”
This is an interesting article, it can be certainly recommended for publication after clarifying some uncertainties.
- Introduction. First sentence. It is important to note that all these compounds are even more important as scintillator materials for different application in high energy physics, medicine and other detector applications. See, for example:
Millers, D.; Grigorjeva, L.; Chernov, S.; Popov, A.; Lecoq, P.; Auffray, E. The temperature dependence of scintillation parameters in PbWO4 crystals. Phys. Status Solidi B 1997, 203, 585–589.
Jung, J.-Y. Luminescent Color-Adjustable Europium and Terbium Co-Doped Strontium Molybdate Phosphors Synthesized at Room Temperature Applied to Flexible Composite for LED Filter. Crystals 2022, 12, 552. https://doi.org/10.3390/cryst12040552
Srivastava, A. M., Brik, M. G., Beers, W. W., Ma, C. G., Piasecki, M., & Cohen, W. E. (2023). Intensity of the Eu3+ hypersensitive transition in isostructural phosphate and vanadate compounds. Journal of Luminescence, 257, 119709.
This is important to attract a wider readership and visibility of this work,
Thanks to the reviewer for the comment. The introduction block is expanded with current information on the applicability of SrMoO4 as follows:
Molybdates with the general formula ABO4 (A = Ba, Sr, Ca) are a class of multifunctional ceramic compounds. SrMoO4 crystallizes in a scheelite structure of tetragonal space group I41/a. SrMoO4 compounds exhibit outstanding luminescent properties and find applications in optical electronics, gas sensors, optical devices, and scintillation detectors [1-12]. Different synthesis and processing techniques lead to changes in structural properties, which affects the material properties and is a motivating factor for the study of SrMoO4 formation.
Several techniques are used to synthesize SrMoO4, including Czochralski method [13], co-precipitation [4,14], traditional solid phase reactions [2,14], sol-gel synthesis [7], hydrothermal synthesis [10], and microwave hydrothermal synthesis [5][12].
- In the introduction, it is important to note how much this ceramic is required for 90Sr immobilization and whether the preparation technique SPS-RS under consideration will cope with this?
We thank the reviewer for the question. Naturally, a scientific experiment is very far from the real conditions of immobilization of any radionuclide. It is not completely clear what amount of the target radionuclide the reviewer has in mind for calculating the amount of ceramics. However, if we make theoretical calculations, we can come to the following conclusions:
1. If we assume that it is necessary to compact 1 ton of radionuclide (assuming that it is 1 ton of pure 90Sr), and taking for immobilization ceramics obtained at 1100 °C, consisting of 27.45% of Sr (Figure 5), we get a figure of 13.717 tons of ceramics.
2. Industrial versions of SPS furnaces can sinter rather large samples. Also such furnaces are presented in the conveyor version, which allows to reduce the time for preparation of each sample [15]. At the moment there are industrial versions of SPS systems of JPX series, capable of producing material with a diameter of up to 300 mm.
Thus, it can be argued that the technology under consideration can cope with the production of oversized samples. However, it should still be taken into account that when using any technology there is a question of not only design and construction and physical principles of operation, but also resource, time and economic costs.
- Whether this will affect the use of this ceramic, its subsequent radiation damage via point defect production? Especially since it is well known that under irradiation point defects are formed in all compounds (SrO, MoO3 and SrMO4-type).
Thanks to the reviewer for his comment. Indeed, according to studies [16], the resistance of ceramics to radiation-induced heating depends directly on the homogeneity of the grain structure. The presence of inhomogeneities and defects accelerates cracking, which leads to deterioration of chemical stability by increasing the contacting reaction surface. However, comparing the hydrolytic stability values of the ceramics obtained at 1100 °C and 1200 °C (Figure 8) as well as the SEM image (Figure 5), it is clear that the low chemical stability of the ceramic sample obtained at 1200 °C is not due to the presence of defects but to the destruction of structural integrity due to sublimation. This makes it clear that the hydrolytic resistance of this type of matrix depends more on structural integrity than on physical integrity.
- Figure 4. It would be useful to perform also Raman measurements
Raman spectra and their description are inserted in the main text of the paper.
- How will porosity affect the declared application?
Thanks to the reviewer for the question. Porosity increases the contact area of the material with the leaching medium, leading to an increase in leaching rate through a large volume of material exposed to surface radionuclide entrainment. However, hydrolytic stability studies and the results reported in the paper suggest that the chemical resistance of matrices is more dependent on structural integrity rather than contact area.
- The list of references should be according to the MDPI standards.
References are corrected to MDPI standard.
In general, the manuscript is interesting and can be recommended for publication after constructive reflection on the above comments.

Reviewer 2 Report
The authors should introduce why spark plasma sintering-reactive sintering strategy would make difference from conventional approaches. Comparison of particle sizes and morphology with literature reported methods should be there.
Possible isomorphous replacement by spark plasma sintering-reactive sintering strategy should be presented.
Author Response
Re1: “SPS-RS technique for SrMoO4 powellite mineral-like ceramic fabrication for 90Sr immobilization” by Belov A.A. and et al. (Manuscript Number: materials-2527932)
Dear, Dear Editors, Dear Reviewers,
We deeply appreciate the time you spent reviewing our paper and the valuable recommendations you made. All the comments are taken into account and corresponding changes are made to the manuscript’s body text. Detailed point-by-point answers are presented below.
Best regards,
Oleg Shichalin, Researcher, Ph.D.
Reviewer #2 comments:
The authors should introduce why spark plasma sintering-reactive sintering strategy would make difference from conventional approaches. Comparison of particle sizes and morphology with literature reported methods should be there.
Thanks to the reviewer for his comment. It is not quite clear what exactly the author means by traditional approaches. Two points of consideration are possible:
- R-SPS as a synthesis approach
- R-SPS as an approach to ceramics formation.
If we consider IPS as a method of ceramics formation, it cannot be compared with taditional synthesis methods, which include the Czochralski method [13], co-precipitation [4,14], traditional solid-phase reactions [2,14], sol-gel synthesis [7], and hydrothermal synthesis [10]. In this perspective, the IPS method can only be compared to compaction technologies, which traditionally include, for example, hot pressing [17].
If we compare IPS from a synthesis point of view, the classical solid-phase reaction [2,13,14,18] may serve as the most suitable comparison. As a result of high temperature treatment and chemical interaction, ceramics of final composition SrMoO4 are formed. Comparing the obtained data by SEM [2,17,18], similar particle morphology as well as particle dimensionality can be observed. However, a significant criterion determining the efficiency of ceramics is their physical characteristics, which is not indicated in the works. In general, the approach using P-SPS technology reduces the number of production stages to 1, since the process of synthesis and formation of ceramics occurs simultaneously, and the total synthesis time is reduced.
Possible isomorphous replacement by spark plasma sintering-reactive sintering strategy should be presented.
Thanks to the reviewer for his comment. The material presented in the paper serves as a proof of the possibility of isomorphic substitution of Ca2+ ions for Sr2+ ions in the crystal structure of CaMoO4. Also, according to Goldschmidt's law, isomorphism is possible under two simultaneous conditions:
1. The sizes of the ions must differ by no more than 10-15%. The ionic radius of Ca2+ is 1.04 Å and the ionic radius of Sr2+ is 1.2, resulting in a difference of 14 %.
2. The difference in electronegativity is less than 0.4. The electronegativity of Ca2+ is 1.00 and that of Sr2+ is 0.95, resulting in a difference of 0.05.
As can be seen, the isomorphism conditions are fulfilled. Additionally, it can be pointed out that the structure of both minerals belongs to the space group I41/a. The analysis of 3d models of the structures presented in the database does not reveal significant distortions of the crystal lattice.
- de Azevedo Marques, A.P.; Umisedo, N.K.; Costa, J.A.; Yoshimura, E.M.; Okuno, E.; Künzel, R. The Role of Capping Agents on the Trapping Levels Structure and Luminescent Emission of SrMoO4 Phosphors. J. Lumin. 2023, 257, 119662, doi:10.1016/j.jlumin.2022.119662.
- Thirmal, C.; Ramarao, S.D.; Rao, L.S.; Murthy, V.R.K. Study of Structural, Dielectric and AC Conductivity Properties of SrMoO4. Mater. Res. Bull. 2022, 146, 111618, doi:10.1016/j.materresbull.2021.111618.
- Mikhaylovskaya, Z.A.; Pankrushina, E.A.; Komleva, E. V.; Ushakov, A. V.; Streltsov, S. V.; Abrahams, I.; Petrova, S.A. Effect of Bi Substitution on the Cationic Vacancy Ordering in SrMoO4-Based Complex Oxides: Structure and Properties. Mater. Sci. Eng. B Solid-State Mater. Adv. Technol. 2022, 281, 115741, doi:10.1016/j.mseb.2022.115741.
- Jung, J. Co-Doped Strontium Molybdate Phosphors Synthesized At. 2022.
- Künzel, R.; Feldhaus, C.M.S.; Suzuki, Y.O.F.; Ferreira, F.F.; de Paula, V.G.; Courrol, L.C.; Umisedo, N.K.; Yoshimura, E.M.; Okuno, E.; de Azevedo Marques, A.P. Photoluminescence and Magnetic Properties of SrMoO4 Phosphors Submitted to Thermal Treatment and Electron Irradiation. J. Magn. Magn. Mater. 2022, 562, 169761, doi:10.1016/j.jmmm.2022.169761.
- Benzineb, M.; Chiker, F.; Khachai, H.; Meradji, H.; Uǧur; Naqib, S.H.; Omran, S. Bin; Wang, X.; Khenata, R. A Comparative Study of Structural, Thermal, and Optoelectronic Properties between Zircon and Scheelite Type Structures in SrMoO4 Compound: An Ab-Initio Study. Optik (Stuttg). 2021, 238, 166714, doi:10.1016/j.ijleo.2021.166714.
- Liu, F.; Wang, J.; Jiang, L.; You, R.; Wang, Q.; Wang, C.; Lin, Z.; Yang, Z.; He, J.; Liu, A.; et al. Compact and Planar Type Rapid Response Ppb-Level SO2 Sensor Based on Stabilized Zirconia and SrMoO4 Sensing Electrode. Sensors Actuators, B Chem. 2020, 307, 127655, doi:10.1016/j.snb.2020.127655.
- Çiftyürek, E.; Sabolsky, K.; Sabolsky, E.M. Molybdenum and Tungsten Oxide Based Gas Sensors for High Temperature Detection of Environmentally Hazardous Sulfur Species. Sensors Actuators, B Chem. 2016, 237, 262–274, doi:10.1016/j.snb.2016.06.071.
- Gao, H.; Yu, C.; Wang, Y.; Wang, S.; Yang, H.; Wang, F.; Tang, S.; Yi, Z.; Li, D. A Novel Photoluminescence Phenomenon in a SrMoO4/SrWO4 Micro/Nano Heterojunction Phosphors Obtained by the Polyacrylamide Gel Method Combined with Low Temperature Calcination Technology. J. Lumin. 2022, 243, 118660, doi:10.1016/j.jlumin.2021.118660.
- Chavan, A.B.; Gawande, A.B.; Gaikwad, V.B.; Jain, G.H.; Deore, M.K. Hydrothermal Synthesis and Luminescence Properties of Dy3+ Doped SrMoO4 Nano-Phosphor. J. Lumin. 2021, 234, 117996, doi:10.1016/j.jlumin.2021.117996.
- Mikhailik, V.B.; Elyashevskyi, Y.; Kraus, H.; Kim, H.J.; Kapustianyk, V.; Panasyuk, M. Temperature Dependence of Scintillation Properties of SrMoO4. Nucl. Instruments Methods Phys. Res. Sect. A Accel. Spectrometers, Detect. Assoc. Equip. 2015, 792, 1–5, doi:10.1016/j.nima.2015.04.018.
- Künzel, R.; Umisedo, N.K.; Okuno, E.; Yoshimura, E.M.; Marques, A.P. de A. Effects of Microwave-Assisted Hydrothermal Treatment and Beta Particles Irradiation on the Thermoluminescence and Optically Stimulated Luminescence of SrMoO4 Powders. Ceram. Int. 2020, 46, 15018–15026, doi:10.1016/j.ceramint.2020.03.032.
- Pankratova, V.; Dunaeva, E.E.; Voronina, I.S.; Kozlova, A.P.; Shendrik, R.; Pankratov, V. Luminescence Properties and Time-Resolved Spectroscopy of Rare-Earth Doped SrMoO4 Single Crystals. Opt. Mater. X 2022, 15, 100169, doi:10.1016/j.omx.2022.100169.
- Ranganatha, C.L.; Lokesha, H.S.; Nagabhushana, K.R.; Palakshamurthy, B.S. Studies on Luminescence Properties of Self-Activated SrMoO4 Phosphor: Kinetic Analysis. J. Alloys Compd. 2023, 962, 171061, doi:10.1016/j.jallcom.2023.171061.
- Tokita, M.; Co, N.J.S. Recent Advanced Spark Plasma Sintering (SPS) Technology , Systems and Applications in Japan; 2013;
- Potanina, E.A.; Orlova, A.I.; Mikhailov, D.A.; Nokhrin, A. V.; Chuvil’deev, V.N.; Boldin, M.S.; Sakharov, N. V.; Lantcev; Tokarev, M.G.; Murashov, A.A. Spark Plasma Sintering of Fine-Grained SrWO4 and NaNd(WO4)2 Tungstates Ceramics with the Scheelite Structure for Nuclear Waste Immobilization. J. Alloys Compd. 2019, 774, 182–190, doi:10.1016/J.JALLCOM.2018.09.348.
- Mikhaylovskaya, Z.A.; Buyanova, E.S.; Malkin, A.I.; Korotkov, A.N.; Knyazev, N.S.; Petrova, S.A. Morphological and Microwave Dielectric Properties of Bi:SrMoO4 Ceramic. J. Solid State Chem. 2022, 316, 1–9, doi:10.1016/j.jssc.2022.123555.
- Tian, X.; Guo, L.; Wen, J.; Zhu, L.; Ji, C.; Huang, Z.; Qiu, H.; Luo, F.; Liu, X.; Li, J.; et al. Anti-Thermal Quenching Behavior of Sm3+ Doped SrMoO4 Phosphor for New Application in Temperature Sensing. J. Alloys Compd. 2023, 959, 170574, doi:10.1016/j.jallcom.2023.170574.

Round 2
Reviewer 1 Report
The authors have successfully improved the original version of their manuscript, responding constructively to all the comments/recommendations of the reviewer. Therefore, the article can be recommended for publication.